# Sustained Cytotoxic Response of Peripheral Blood Mononuclear Cells from Unvaccinated Individuals Admitted to the ICU Due to Critical COVID-19 Is Essential to Avoid a Fatal Outcome

**DOI:** 10.3390/ijerph20031947

**Published:** 2023-01-20

**Authors:** Guiomar Casado-Fernández, Magdalena Corona, Montserrat Torres, Adolfo J. Saez, Fernando Ramos-Martín, Mario Manzanares, Lorena Vigón, Elena Mateos, Francisco Pozo, Inmaculada Casas, Valentín García-Gutierrez, Sara Rodríguez-Mora, Mayte Coiras

**Affiliations:** 1Immunopathology Unit, National Center of Microbiology, Instituto de Salud Carlos III, 28029 Madrid, Spain; 2Faculty of Sciences, Universidad de Alcalá, 28805 Madrid, Spain; 3Biomedical Research Center Network in Infectious Diseases (CIBERINFEC), Instituto de Salud Carlos III, 28220 Madrid, Spain; 4Hematology and Hemotherapy Service, Instituto Ramón y Cajal de Investigación Sanitaria (IRYCIS), Hospital Universitario Ramón y Cajal, 28034 Madrid, Spain; 5Respiratory Viruses Service, National Center of Microbiology, Instituto de Salud Carlos III, 28029 Madrid, Spain

**Keywords:** SARS-CoV-2, COVID-19, immune response, cytotoxic response, IL-15

## Abstract

The main objective of this study was to determine the influence of the cytotoxic activity of peripheral blood mononuclear cells (PBMCs) on the outcome of unvaccinated individuals with critical COVID-19 admitted to the ICU. Blood samples from 23 individuals were collected upon admission and then every 2 weeks for 13 weeks until death (Exitus group) (*n* = 13) or discharge (Survival group) (*n* = 10). We did not find significant differences between groups in sociodemographic, clinical, or biochemical data that may influence the fatal outcome. However, direct cellular cytotoxicity of PBMCs from individuals of the Exitus group against pseudotyped SARS-CoV-2-infected Vero E6 cells was significantly reduced upon admission (−2.69-fold; *p* = 0.0234) and after 4 weeks at the ICU (−5.58-fold; *p* = 0.0290), in comparison with individuals who survived, and it did not improve during hospitalization. In vitro treatment with IL-15 of these cells did not restore an effective cytotoxicity at any time point until the fatal outcome, and an increased expression of immune exhaustion markers was observed in NKT, CD4+, and CD8+ T cells. However, IL-15 treatment of PBMCs from individuals of the Survival group significantly increased cytotoxicity at Week 4 (6.18-fold; *p* = 0.0303). Consequently, immunomodulatory treatments that may overcome immune exhaustion and induce sustained, efficient cytotoxic activity could be essential for survival during hospitalization due to critical COVID-19.

## 1. Introduction

Several cases of atypical viral pneumonia related to an outbreak in a seafood market in Wuhan City (Hubei Province, China) were detected in December 2019. A novel coronavirus termed severe acute respiratory syndrome coronavirus 2 (SARS-CoV-2) rapidly spread across the world causing the pandemic of coronavirus disease 2019 (COVID-19) [1]. Since then, about 626 million cases of COVID-19 and 6.5 million associated deaths have been reported around the world (updated October 2022) [2].

Different risk factors and comorbidities may influence the progression of COVID-19 among individuals, which affected the comparison of clinical data between the affected countries from the beginning of the pandemic [3]. However, it was determined that the clinical manifestations of COVID-19 may vary from asymptomatic infection to acute respiratory distress syndrome (ARDS) [4]. Among the different risk factors that were associated with a higher probability to develop a severe form of COVID-19 are old age, male gender, and previous comorbidities such as diabetes, hypertension, and/or obesity [5,6]. Severe and critical COVID-19 have been also associated with cytopenia, mostly of CD4+ T lymphocytes, as well as with excessive exhaustion of natural killer (NK) and CD8+ T cells, which results in an immunocompromised state that is unable to clear the infection, in addition to the development of serious complications such as cytokine storm and thrombotic events [7].

The main cause for developing a severe form of the disease and death in patients with COVID-19 has been considered an excessive inflammatory response that may promote the development of shock and/or hypercytokinemia with multiorgan dysfunction [4]. This syndrome is known as cytokine storm [8], and it is similar to the hemophagocytic lymphohistiocytosis (sHLH) disease, which consists of a state of hyperinflammation triggered by a viral infection [9,10]. It can be related to the fact that SARS-CoV-2 uses the receptor of angiotensin-converting enzyme 2 (ACE2) to infect the target cells. The binding of SARS-CoV-2 to the ACE2 receptor, which is widely expressed by cells of different tissues [11], leads to an increase in angiotensin II, which activates the nuclear factor-κB (NF-κB). The activation of this essential transcription factor stimulates the expression of proinflammatory cytokines, chemokines, and adhesion molecules [12]. Despite this excessive inflammatory response, it seems to be ineffectual to clear the infection and, due to T-cell activation, may end in apoptosis [13]. This cytokine storm can be responsible for lymphopenia [12,14] and also decreased levels of NK and CD8+ T cells with an exhausted phenotype and an increased expression of inhibitory receptors such as NKG2A in NK cells and PD-1 in CD8+ T cells [15,16]. Moreover, the expression of degranulation markers such as CD107a is usually reduced on the surface of these cells [15], proving the existence of an impairment in the capacity to eliminate infected cells in individuals with severe and critical COVID-19 [17].

Despite the rapid global response to develop and administer effective vaccines, the emergence of new variants that may escape from the vaccine-induced immune response has resulted in a rapid decline in the protection against symptomatic illness [18]. Consequently, it is necessary to continue the research for efficient therapies against COVID-19. Several treatments have been tested to improve the status of individuals hospitalized due to severe and critical disease, but none has been proven yet to be specific to eliminating the infected cells or reducing the infection rate [15,19]. Therefore, supportive treatment such as oxygen and fluid therapy is still among the most important strategies applied for the management of these individuals [2,4,16,20]. Unspecific treatments such as corticosteroids have also been beneficial to control and reduce the excessive inflammatory response [21,22]. Recently, several drugs have been authorized by FDA for the treatment of individuals with mild-to-moderate COVID-19, such as molnupiravir, nirmatrelvir, and ritonavir, but only in case of emergency for those individuals with a high risk to progress to severe disease [23,24,25,26]. Molnupiravir shows a broad-spectrum antiviral activity due to its capacity to induce RNA mutagenesis by the interference with several viral RNA-dependent RNA polymerases (RdRp), including the RdRp used by SARS-CoV-2 for replication and transcription of its RNA genome [27]. The combination of nirmatrelvir and ritonavir has also demonstrated to be able to stop the spread of the infection in animal models by inhibiting the main viral protease 3CLpro of SARS-CoV-2 [28]. However, these antiviral drugs need to be administered in conjunction with other measures such as vaccines in order to trigger a potent cytotoxic immune response able to eliminate the infected cells.

Most cells with cytotoxic activity, such as CD8+ T lymphocytes and NK and NKT cells, require the presence of interleukin-15 (IL-15) for their function and homeostatic regulation [29]. IL-15 primes CD8+ T cells for their activation by specific antigens and also improves NK cell cytotoxicity and proliferation. Therefore, IL-15 may enhance both innate and adaptive cellular immune responses against the infected cells [22,25]. IL-15 also prevents NK and T cells from apoptosis by upregulating anti-apoptotic factors such as Bcl-2 and downregulating pro-apoptotic factors such as GSK-3 [26], thereby avoiding cytopenia. Accordingly, IL-15 has been proposed as a novel immunomodulatory cytokine in cancer immunotherapy and has been considered an adjuvant in vaccine regimens [26,30]. Phase 1 clinical trials with human bolus i.v. infusion IL-15 (rhIL-15) or IL-15 superagonist N-803 have demonstrated that this cytokine may induce significant expansion and/or activation of CD4+, CD8+, and NK effector cells in vivo, and therefore, it has been applied to treat cancer or viral infections such as HIV [31,32,33,34,35]. Therefore, the use of immunomodulatory agents such as IL-15 that may restore the cytotoxic activity in individuals hospitalized due to critical COVID-19 could be useful to help clear the virus from the organism.

In this observational, longitudinal study, we characterized the evolution of the cytotoxic immune response against SARS-CoV-2 in peripheral blood mononuclear cells (PBMCs) isolated from individuals diagnosed with COVID-19 who were admitted to the intensive care unit (ICU) due to severe complications, as well as the influence of this response on the final outcome in comparison with other essential risk factors. We also evaluated the capacity of these cells to respond to treatment with IL-15 and enhance their cytotoxic antiviral response. The results obtained in this study may contribute to a better understanding of the role of the cytotoxic response in the fatal outcome of COVID-19 and to advance toward the development of effective immunomodulatory treatments that promote viral clearance.

## 2. Materials and Methods

### 2.1. Study Subjects

Twenty-three individuals with critical COVID-19 admitted to the ICU of Hospital Universitario Ramón y Cajal (Madrid, Spain) from October 2020 until April 2021 were recruited for this study. This period covered both the second and third pandemic waves of COVID-19 in Spain (from June to December 2020 and from December 2020 to March 2021, respectively). The participants were randomly recruited upon hospitalization according to the following criteria: authorization of informed written or oral witnessed consent to participate in the study, hospitalized at the ICU due to critical COVID-19, positive SARS-CoV-2 RT-qPCR assay in nasopharyngeal smear performed at the hospital admission according to internal validated protocols, older than 18 years old, and unvaccinated against SARS-CoV-2 at time of infection. Peripheral blood samples and clinical data were collected every 2 weeks for a total of 13 weeks, and blood samples were processed and cryopreserved until the moment of analysis. This period was calculated taking into consideration that the length of hospital stay in Spain during the first pandemic waves was estimated in 35 days on average [36], so we could perform at least two complete rounds of recruitment and follow-up. After 13 weeks, the participants were divided into two groups according to the final outcome: fatal outcome (*n* = 13), henceforth Exitus; or hospital discharge (*n* = 10), henceforth Survival.

### 2.2. Ethical Statement

The individuals who participated in this study were recruited from Hospital Universitario Ramón y Cajal (Madrid, Spain). All of them gave informed written consent to participate in the study or witnessed oral consent with written consent by a representative to avoid handling contaminated documents. Current Spanish and European Data Protection Acts ensured the confidentiality and anonymity of all participants. Protocol for this study (CEI PI 32_2020-v2) was prepared in accordance with the Helsinki Declaration and previously reviewed and approved by the Ethics Committees of Instituto de Salud Carlos III (IRB IORG0006384) and the participating hospital.

### 2.3. Cells

Five milliliters of whole blood was collected in EDTA Vacutainer tubes (Becton Dickinson, Madrid, Spain) and immediately processed to isolate PMBCs and plasma by Ficoll-Hypaque (Pharmacia Corporation, North Peapack, NJ, USA) density gradient centrifugation, and then they were cryopreserved until the analysis. Cell viability after thawing was assessed by optical microscopy and flow cytometry. The Vero E6 (African green monkey kidney) cell line (ECACC 85020206) was kindly provided by Dr. Antonio Alcami (CBM Severo Ochoa, Madrid, Spain), and these cells were cultured in DMEM supplemented with 10% FCS, 2 mM L-glutamine, and 100 units/mL penicillin and streptomycin (Lonza, Basel, Switzerland).

### 2.4. Pseudotyped SARS-CoV-2 Infection Assay

Cytotoxic activity against SARS-CoV-2 of PBMCs from the participants was analyzed using Vero E6 cells infected by single-cycle pseudotyped virus pNL4-3Δenv_SARS-CoV-2-SΔ19(G614)_Ren. This virus encodes SARS-CoV-2 spike glycoprotein with mutation D614G within the HIV-1 genome, as well as the Renilla luciferase gene generated as previously described [37,38]. Variants of SARS-CoV-2 containing the D614G mutation in the spike (S) protein began to circulate early in the pandemic and became rapidly dominant in many regions [39,40]. Briefly, a monolayer of Vero E6 was infected with pNL4-3Δenv_ SARS-CoV-2-SΔ19(G614)_Ren (100 ng p24 Gag/well) for 48 h. Then cells were washed and co-cultured for 1 h with PBMCs from the participants (ratio 1:1), previously treated or not with IL-15 (13 ng/mL) for 48 h. The dose of IL-15 used for this study was selected, taking into account the dose previously described that causes the highest effect on NK proliferation (10–25 ng/mL) [41] and also the preferential expansion of memory cytotoxic CD8+ T cells (≤5 ng/mL) [42]. The cytotoxic activity of the PBMCs on the monolayer of Vero E6 was determined after removal of the supernatant with the cells in suspension and dissociation of Vero E6 cells from the plate with trypsin-EDTA solution (Sigma Aldrich-Merck, Darmstadt, Germany) and subsequent quantification of caspase-3 activity in the detached cells by luminescence using Caspase-Glo 3/7 Analysis system (Promega, Madison, WI, USA). In order to determine which cytotoxic cell populations were responsible for this activity, PBMCs were collected from the supernatant and then analyzed by flow cytometry.

### 2.5. Flow Cytometry Analysis

For staining of cell surface phenotyping markers, the following conjugated antibodies were used: CD3-PE, CD8-APC-H7, CD56-FITC, and CD107a-PE-Cy7, purchased from BD Biosciences (San Jose, CA, USA). For staining of exhaustion markers, the following conjugated antibodies were used: CD3-BV510, CD8-APC-H7, CD56-FITC, and PD1-BV650, purchased from BD Biosciences, as well as CD4-PE, purchased from Immunostep S.L. (Salamanca, Spain) and TIGIT-AlexaFluor700 purchased from Thermo Fisher (Waltham, MA, USA). Samples were acquired by using BD LSRFortessa X-20 flow cytometer with FACS Diva software v 6.0 (BD Biosciences, Franklin Lakes, NJ, USA) and then analyzed using FlowJo software v10.0.7 (Tree Star Inc., Ashland, OR, USA).

### 2.6. Detection of SARS-CoV-2 RNAemia

Total RNA was extracted from plasma samples using QIAamp MinElute Virus Kit (Qiagen Iberia, Madrid, Spain) in an automated extractor QIAcube (Qiagen, Hilden, Germany). SARS-CoV-2 RNA detection was performed by RT-qPCR assay with targets in the envelope (E) and nucleocapsid (N) genes, as previously described [43], which is part of the Interim Guidance of the World Health Organization (WHO) for the diagnostic testing of SARS-CoV-2 [44]. Samples were considered positive for the analysis when the quantification cycle (Cq) value was below 45 cycles.

### 2.7. Statistical Analysis

All statistical analyses and graphics were performed using GraphPad Prism software version 8.4.3 (GraphPad Software Inc., San Diego, CA, USA). Group comparisons were performed using Mann–Whitney U-test, Wilcoxon matched-pairs signed rank test, or one-way ANOVA and Tukey’s multiple comparisons test. Fisher’s exact test was used to evaluate differences in the clinical characteristics between the two groups. Values of *p* < 0.05 (two-tailed) were considered statistically significant. The mean and standard deviation were used to calculate Cohen’s d statistic by using an Excel effect size calculator [45]. An effect size of 0.2, 0.5, 0.8, or ≥1.2 was an indication of small, medium, large, and very large effect sizes, respectively [46].

## 3. Results

### 3.1. Characteristics of the Participants

This observational, longitudinal study included 23 patients diagnosed with COVID-19 who were admitted to the ICU due to severe complications. Blood samples were collected every 2 weeks for a total of 13 weeks or until the final outcome. None of the participants in this study was reported to be vaccinated against COVID-19 at the time of infection and hospitalization. The main sociodemographic and clinical characteristics of all participants are summarized in Table 1.

The median age of the Exitus group was 65.0 years (IQR 62.0–69.0), whereas the median age of the Survival group was 63.0 years (IQR 59.0–68.5). Most participants (73.9%) were men, and 65.2% had at least one comorbidity. The main comorbidities were dyslipidemia (52.2%), hypertension (39.1%), and diabetes mellitus (26.1%). Most patients received corticoids (82.6%), antibiotics (82.6%), and anticoagulants (82.6%) as standard treatment. They also received other treatments such as tocilizumab (17.4%) and remdesivir (8.7%). The most common signs and symptoms were pneumonia (100%), cough and expectoration (87.0%), dyspnea (87.0%), and fever (87.0%). During their stay at the ICU, 12 patients (92.3%) from the Exitus group and 10 patients (100%) from the Survival group required invasive mechanical ventilation. We did not find significant differences between both groups in the sociodemographic and clinical data. We also analyzed blood biochemistry data at baseline (first sample) between both groups, and we did not find significant differences between groups either. More detailed information about the clinical data, hospitalization, and blood biochemistry data of every participant is shown in Appendix A.

### 3.2. Length of Hospital and ICU Stay 

The median length of hospital stay was 59.0 (IQR 36.0–100.0) days in the Exitus group and 73.0 (IQR 44.5–90.0) days in the Survival group (Figure 1A). The median length of ICU stay was 49.0 (IQR 25.5–82.0) days in the Exitus group and 44.0 (IQR 25.0–66.5) days in the Survival group (Figure 1B). No significant differences in these parameters were found between both groups.

### 3.3. Levels of Peripheral Blood Lymphocytes

Individuals from the Exitus group showed lymphopenia (<1000 cells/mm^3^) only at the time of hospitalization (t = 0), but at Week 2, all participants showed levels of lymphocytes above the threshold for lymphopenia (Figure 2). Lymphocyte count was 1.54 (*p* = 0.0318; Cohen’s d = 1.00)- and 1.71 (*p* = 0.0030; Cohen’s d = 3.17)-fold higher in patients from the Survival group in comparison with the Exitus group at baseline and after 4 weeks of hospitalization, respectively. Although both groups showed an increase in the levels of lymphocytes in peripheral blood during hospitalization, they remained on average 1.54-fold higher in individuals of the Survival group in comparison with the participants of the Exitus group for at least 10 weeks. In the Survival group, the lymphocyte counts steadily increased until Week 6 of hospitalization and then decreased until Week 10.

### 3.4. PBMCs Cytotoxic Activity against Vero E6 Cells Infected with Pseudotyped SARS-CoV-2

We evaluated the cytotoxic activity of PBMCs of the hospitalized patients in the longitudinal samples by quantifying caspase-3 activation in a monolayer of Vero E6 cells infected with pseudotyped SARS-CoV-2 after co-culture for 1 h (ratio 1:1). The cytotoxic capacity of PBMCs isolated from individuals of the Exitus group did not significantly change during the hospitalization, and it remained steadily low until the fatal outcome (Figure 3). This cytotoxic activity did not significantly increase when these cells were stimulated with IL-15 for 48 h. In the Survival group, the cytotoxic activity of PBMCs was increased 2.69-fold (*p* = 0.0234; Cohen’s d = 1.12) in the basal sample (t = 0) in comparison with the Exitus group, and it steadily increased until Week 6 of hospitalization. This increase showed statistical significance in the comparison between groups at Week 4 (5.58-fold; *p* = 0.0290; Cohen’s d = 1.28). From Week 6 onwards, the cytotoxic activity decreased in the Survival group until hospital discharge, which was in accordance with the decay of the lymphocyte count in the same week of hospitalization (see Figure 2). Treatment with IL-15 increased 2.30-fold on average the cytotoxic activity of PBMCs of individuals from the Survival group in all samples during the follow-up. After 4 weeks of hospitalization, IL-15-induced cytotoxic activity was increased 6.18-fold (*p* = 0.0303; Cohen’s d = 1.05) in the PBMCs of the individuals from the Survival group.

### 3.5. Levels of NK and NKT Cells

The levels of cells with cytotoxic phenotypes were analyzed in the PBMCs that were co-cultured with Vero E6 cells infected with pseudotyped SARS-CoV-2. We did not find significant differences between the levels of cells with NK phenotype (CD3-CD56+) between both groups of individuals, even after treatment with IL-15 (Figure 4A). The expression of the degranulation marker CD107a on the surface of these cells remained unchanged in the individuals from the Exitus group. Although this expression steadily increased from Week 4 until hospital discharge in the Survival group, this difference did not achieve statistical significance in the comparison with the Exitus group (Figure 4B). Treatment with IL-15 did not significantly modify the expression of CD107a in these cells.

From Week 4, PBMCs from individuals of both groups showed an increase in the levels of cells with NKT phenotype (CD3+CD56+) that was 4.17-fold higher in the PBMCs from the individuals of the Exitus group at Week 6, although this difference did not achieve significance (*p* = 0.0571; Cohen’s d = 3.15) (Figure 5A). This increase was slightly higher when cells were treated with IL-15 in both groups, although data did not reach statistical significance either (Figure 5A). The expression of CD107a did not significantly change between both groups or after treatment with IL-15 (Figure 5B).

### 3.6. Levels of CD8+ T Cells

CD8 count was enhanced 2.07-fold (*p* = 0.0409; Cohen’s d = 0.94) in the basal sample (t = 0) of the Exitus group in comparison with the Survival group, and these levels remained steady until the fatal outcome (Figure 6A). IL-15 did not produce a significant beneficial effect on the levels of this cell population. In the Survival group, CD8+ T cells were reduced 1.51-fold on average in comparison with the Exitus group (Figure 6A). The expression of CD107a decreased 1.53-fold (*p* = 0.0313) at Week 4 in CD8+ T cells from the individuals of the Exitus group, and it remained unchanged until the fatal outcome. Treatment with IL-15 showed a slight benefit on the expression of this degranulation marker, but this difference was not significant (Figure 6B). In the Survival group, there was also a decrease in the expression of CD107a in CD8+ T cells at Week 4, and then there was an increase onwards, but no significance was achieved before or after treatment with IL-15 (Figure 6B).

### 3.7. Levels of Exhaustion Markers

The expression levels of the immune exhaustion markers PD-1 (Programmed cell death protein) and TIGIT (T cell immunoreceptor with Ig and ITIM domains) were analyzed by flow cytometry in CD4+ T cells and cytotoxic cells from all participants. Samples obtained at the time of hospital admission (t = 0) until Week 6 of hospitalization were analyzed, due to the lack of samples available after this time. CD4+ and CD8+ T cells from individuals of the Exitus group showed levels of PD-1 that were significantly increased 1.72 (*p* = 0.0095; Cohen’s d = 6.87)- and 2.92-fold (*p* = 0.0317; Cohen’s d = 2.07) after 4 weeks of hospitalization, respectively, in comparison with individuals from the Survival group (Figure 7A,B, left graphs). In NKT cells, the levels of TIGIT significantly increased from the time of hospital admission until 6 weeks of hospitalization in individuals from the Exitus group, and these levels were increased 1.76-fold (*p* = 0.0357; Cohen’s d = 0.33) after 2 weeks of hospitalization, in comparison with the Survival group (Figure 7C, right graph). No significant differences between groups were observed in the expression levels of these exhaustion markers in NK cells (Figure 7D).

### 3.8. Quantification of SARS-CoV-2 RNAemia in Plasma

The presence of RNA from SARS-CoV-2 was analyzed in the plasma of all individuals in the basal sample (t = 0), and no positive results were obtained in any participant, with Cq values below the limit of positive detection of 45 cycles.

## 4. Discussion

SARS-CoV-2 infection may cause several clinical presentations of COVID-19 that range from asymptomatic infection to severe, critical, or even fatal disease [4]. Although the general vaccination of the population has greatly reduced the severe forms of the disease, the emergence of new variants of SARS-CoV-2 is increasing again the number of individuals who develop critical COVID-19 and need to be hospitalized due to breakthrough infections [47,48]. Therefore, it is still a priority to develop preventive and therapeutic strategies that may avoid the development of critical forms of COVID-19 and a fatal outcome in hospitalized patients.

It has been described that older people with comorbidities present a higher risk of developing severe and critical COVID-19 [49]. In Spain, the most significant independent predictors of death due to COVID-19 are being male over 50 years old with associated comorbidities such as hypertension and diabetes [36]. In our study, most of the participants of both groups presented these characteristics, without significant differences between them in sociodemographic, clinical, or biochemical data that may influence the disease severity or the fatal outcome. In fact, both groups showed increased levels of biochemical parameters that have been associated with a higher risk of a poor outcome of COVID-19, such as C-reactive protein (CRP) and lactate dehydrogenase (LDH) [50,51,52,53,54,55,56]. On the other hand, although it has been reported that individuals with severe COVID-19 may have detectable SARS-CoV-2 RNAnemia [54], which has been related to an increased risk of mortality [55], none of the participants in our study presented detectable RNA in serum, so we could not establish a correlation between detectable viremia and a fatal outcome. Finally, COVID-19 shows a higher length of hospital stay and mortality rate in comparison with other infectious diseases that require ICU admissions, such as community-acquired pneumonia (CAP) or sepsis [56], and it was estimated in 35 days on average for those individuals who were hospitalized in Spain during the first pandemic waves [36]. In our study, the median length of hospital stay was similar between both the Exitus and Survival groups (59.0 and 73.0 days, respectively), and 50% of all participants stayed more than 40 days in the ICU. Therefore, other differential factors have to be considered to explain the fatal outcome in the Exitus group and the recovery in the Survival group.

Lymphopenia is also considered a risk factor for disease severity that has been related to a longer length of hospital stay [57]. In our study, 73.9% of the participants of both the Exitus and Survival groups showed low lymphocyte counts. However, the level of total lymphocytes was higher on average in the participants of the Survival group, in comparison with the Exitus group. This is in accordance with the previous observation that those individuals who died due to COVID-19 develop a more severe lymphopenia over time in comparison to those who survive [58], and that these lymphocytes usually show an exhausted phenotype, which results in a gradual immunodeficiency [15,16]. Due to an effective cytotoxic antiviral response being essential for the control of viral infections, including SARS-CoV-2 [17], an immune reconstitution is crucial for the recovery from COVID-19. The acceleration of this immune reconstitution greatly depends on the homeostatic peripheral expansion of lymphocytes, which could be a good strategy to improve the outcome of individuals hospitalized due to severe COVID-19 [58].

IL-15 is a cytokine necessary for the function and homeostasis of CD8+ T cells, NK and NKT cells [29], and it has been proposed as a novel immunomodulatory treatment for several diseases such as cancer and HIV infection [26,30,35,59]. Therefore, in this study, we evaluated whether treatment in vitro with IL-15 could improve the cytotoxic response in PBMCs of hospitalized individuals with severe and critical COVID-19 as the infection progressed, in order to determine its validity as a candidate to stimulate the immune reconstitution during COVID-19. With the dose assayed, we observed an overall increase in the cytotoxic response against target cells infected with pseudotyped SARS-CoV-2 in the PBMCs of those individuals who survived COVID-19 after 44 days on average of hospitalization at the ICU. This cytotoxic response was effectively increased 2.3-fold on average upon treatment with IL-15 in all samples collected during the follow-up, with Week 6 of hospitalization being the turning point from which both the lymphocyte count and the cytotoxic activity began to decrease until hospital discharge. Despite sharing similar sociodemographic, clinical, and biochemical characteristics and having been hospitalized for nearly the same amount of time (59 days in total, of which 49 days were at the ICU), the group of individuals who did not survive COVID-19 showed PBMCs that were unresponsive in the presence of the infected target cells, even when they were stimulated with IL-15. The total levels of lymphocytes were above the threshold of lymphopenia in the individuals from this group since Week 2 of hospitalization. However, these levels did not significantly increase during the time of hospitalization, and no effective antiviral response was detected, even in the presence of IL-15 at the assayed dose. The absence of response to IL-15 has been described for other chronic infections such as human immunodeficiency virus (HIV), hepatitis C virus (HCV), hepatitis B virus (HBV), and human T lymphotropic virus (HTLV), in which the immune exhaustion may affect the expression of the specific receptor of IL-15, CD122, in CD8+ T cells [60], making them unresponsive to this homeostatic stimulus. The immune exhaustion that is characteristic of COVID-19 has not only been observed in CD8+ T cells but also in other cytotoxic cell types such as NK and NKT cells [15,16]. Therefore, the absence of response to IL-15 in PBMCs from the Exitus group might be related to an increased immune exhaustion that could not be overcome by IL-15 treatment alone at the assayed dose. This has also been observed in Phase I clinical trials in which monotherapy with IL-15 was ineffective despite the huge expansion of cytotoxic cells [31,32]. Consequently, other costimulatory treatments would be necessary to induce an effective response in these individuals, such as immunological checkpoint inhibitors or monoclonal antibodies targeting the infected cells, to enhance the antibody-dependent cellular cytotoxicity (ADCC).

We also determined the cell types that were mainly involved in the development of this enhanced cytotoxic response in the group of individuals who survived COVID-19. It has been demonstrated that the presence of functional CD8+ T cells is essential to protect animal models from the development of severe COVID-19 [61,62]. Moreover, the absence of an effective CD8-mediated cytotoxic response has been appointed as an essential factor for the increased susceptibility of elderly people to develop severe or critical COVID-19 [63]. Surprisingly, the levels of CD8+ T cells were 2.07-fold (*p* = 0.0409) higher in the Exitus group at the time of hospitalization (t = 0), and they remained enhanced until the fatal outcome, in comparison with the Survival group. Although the individuals from the Exitus group showed lymphocyte counts slightly above the threshold of lymphopenia, these levels remained nearly unchanged throughout the time they were hospitalized. Moreover, it has been described that lymphopenia may lead to a pattern of coexisting suppression and activation in which there is a peripheral loss of T cells concurrently with intense proliferation of the CD8+ T cell pool [64,65]. This would account for the slightly increased levels of CD8+ T cells observed in the PBMCs of the participants from the Exitus group since the Week 6 of hospitalization onwards, despite the total levels of lymphocytes remaining unchanged and above the threshold of lymphopenia, also indicating a potential reduction of CD4+ T cells. However, these CD8+ T cells showed low cytotoxic activity, as was determined by the low expression of the degranulation marker CD107a on the cell surface, whereas the expression of this marker increased in CD8+ T cells from the participants of the Survival group from Week 4 onwards. The expression of CD107a is considered a biomarker to evaluate the cytotoxic capacity [66], and it has been described to be reduced in the surface of cells isolated from individuals with COVID-19, in comparison with uninfected, healthy donors [15]. Therefore, the increased expression of CD107a on the surface of CD8+ T cells may at least partially explain the contribution of these cells to the enhanced cytotoxic activity observed in PBMCs of the participants from the Survival group. In addition, we observed an increased expression of the exhaustion marker PD-1 in both CD4+ and CD8+ T cells from individuals of the Exitus group, which may also contribute to the reduced cytotoxic activity and response to IL-15 of these cells.

Other cell populations may be involved in the cytotoxic activity against the pseudotyped SARS-CoV-2-infected cells, such as NK and NKT cells. As occurs with other severe acute infections, NK cells may be recruited to the site of infection with SARS-CoV-2 [67], which would explain the reduced numbers of circulating NK cells that were observed in all participants. Moreover, the cytokine storm that is developed during severe and critical COVID-19 might result in a dysfunctional status of NK cells that would express high levels of inhibitory checkpoint receptors such as NKG2A or PD-1 [68]. In our study, the levels of NK cells were similar between both groups until Week 6, in which the participants of the Survival group showed a progressive increase until hospital discharge. This correlated with an increased expression of CD107a, which has been associated in NK cells with an enhanced cytokine secretion and improved lysis of target cells [7,69], although the percentage of NK cells expressing activation markers such as CD107a has been described to be lower in individuals with COVID-19 in comparison to healthy controls [70]. Conversely, although the levels of NKT cells were greatly enhanced in the Survival group from Week 6 onwards, the expression of CD107a remained nearly unchanged, while they showed enhanced levels of the exhaustion marker TIGIT on the cell surface that steadily increased until Week 6 of hospitalization.

One of the most important limitations of our study was the reduced sample size. Nevertheless, statistical significance was achieved between groups in several essential parameters that also showed a large or very large effect magnitude when the pooled differences between groups were calculated.

## 5. Conclusions

In conclusion, the most important contribution of this study to the fight against the immunopathology caused by SARS-CoV-2 infection is the confirmation that the impaired antiviral capacity of PBMCs appeared to be a critical factor in the progression to fatal COVID-19, even above other sociodemographic, clinical, and biochemical factors. Therefore, the use of immunomodulatory treatments that may enhance cytotoxicity may contribute to a positive outcome during hospitalization due to critical COVID-19. The increased cytotoxic activity mediated by IL-15 in individuals from the Survival group did not likely rely on increased levels of cytotoxic cells but on the immunocompetency of these cells to respond to such stimulus. Therefore, the monitorization of the expression of exhaustion markers and the responsiveness to homeostatic stimuli such as IL-15 of these cell populations in hospitalized individuals could be useful as a predictor marker of a fatal outcome in order to initiate additional measures to prevent it. Consequently, this observational study is a proof of concept that the use of IL-15 in individuals hospitalized with COVID-19 could improve the antiviral immune response, although this stimulus alone would not be effective in those individuals with unresponsive cells due to immune exhaustion or impaired IL-15R signaling pathway, which is essential in the prevention of exhaustion [71], even when they show lymphocyte counts above the threshold of lymphopenia. More studies with a larger population are necessary to evaluate combinations of increasing doses of IL-15 with other immunomodulatory agents such as immune checkpoint inhibitors or monoclonal antibodies targeting the infected cells to help restore and enhance the antiviral response during critical forms of COVID-19.

## Figures and Tables

**Figure 1 ijerph-20-01947-f001:**
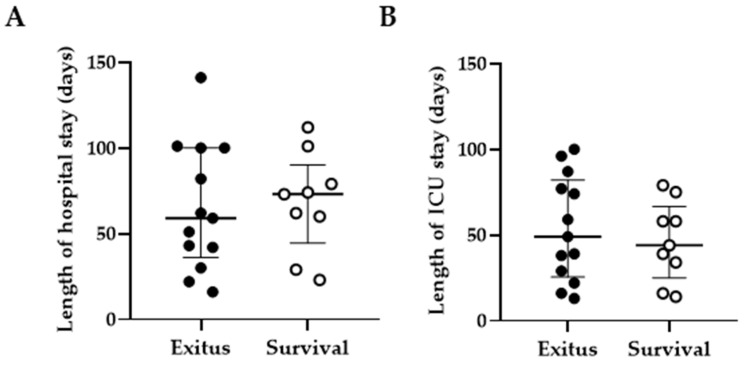
Length of hospital and ICU stay. Length of hospital (**A**) and ICU (**B**) stay (days) in individuals with critical COVID-19 who were divided into two groups according to the final outcome: fatal outcome, Exitus group (filled dots); or recovery and hospital discharge, Survival group (blank dots). Each dot corresponds to one individual, and vertical lines represent mean ± standard error of the mean (SEM). Statistical significance between groups was calculated using Mann–Whitney U-test.

**Figure 2 ijerph-20-01947-f002:**
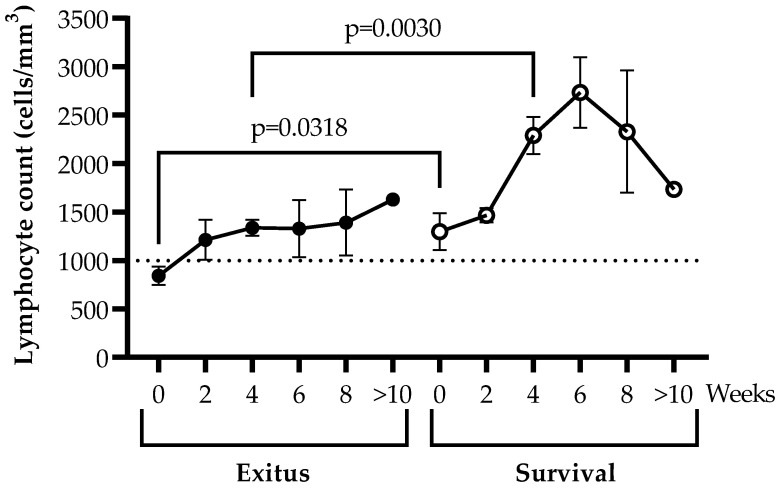
Peripheral blood lymphocyte counts during hospitalization due to critical COVID-19. Lymphocyte count (cells/mm^3^) was determined every 2 weeks for a total of 13 weeks or until the final outcome in blood samples from the participants of the Exitus (filled dots) and Survival (blank dots) groups. The threshold for lymphopenia is shown at 1000 cells/mm^3^ with a dotted line. Each dot corresponds to the mean of data, and vertical lines represent SEM. Statistical significance between groups was calculated using Mann–Whitney U-test, and within groups, Wilcoxon U-test was used.

**Figure 3 ijerph-20-01947-f003:**
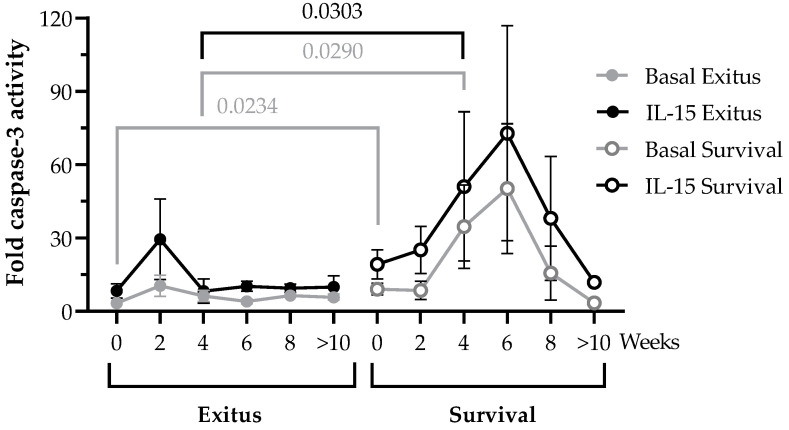
Direct cellular cytotoxicity of PBMCs isolated from hospitalized patients with critical COVID-19. Cytotoxic capacity of PBMCs from individuals of the Exitus (filled dots) and Survival (blank dots) groups measured by quantifying caspase-3 activity in a monolayer of Vero E6 cells infected with pseudotyped SARS-CoV-2 after co-culture (1:1) for 1 h in basal conditions (gray lines) or after treatment with IL-15 for 48 h (black lines). Each dot corresponds to the mean of data, and vertical lines represent SEM. Statistical significance between groups was calculated using Mann–Whitney U-test, and within groups, Wilcoxon U-test was used.

**Figure 4 ijerph-20-01947-f004:**
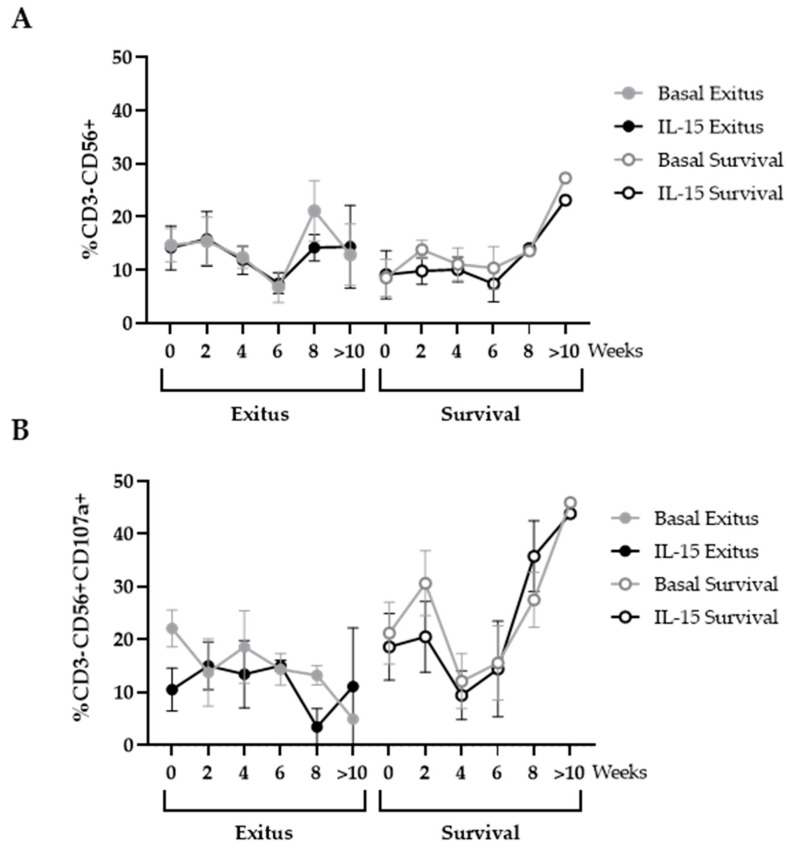
Levels of NK cells in PBMCs from individuals hospitalized with critical COVID-19. (**A**) Total levels of NK cells (CD3-CD56+) and (**B**) expression of the degranulation marker CD107a in these cells was evaluated in PBMCs in basal conditions (gray lines) or after treatment with IL-15 for 48 h (black lines) after co-culture (1:1) with Vero E6 cells infected with pseudotyped SARS-CoV-2 for 1 h. Each dot corresponds to the mean of data, and vertical lines represent SEM. Statistical significance between groups was calculated using Mann–Whitney U-test, and within groups, Wilcoxon U-test was used.

**Figure 5 ijerph-20-01947-f005:**
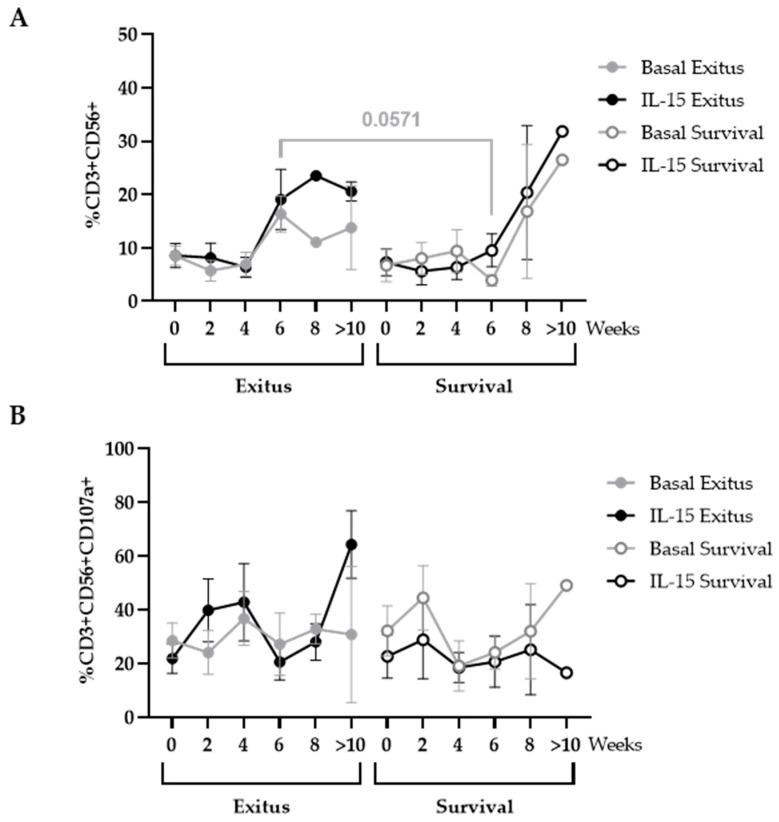
Levels of NKT cells in PBMCs from individuals hospitalized with critical COVID-19. Total levels of NKT cells (CD3+CD56+) (**A**) and expression of the degranulation marker CD107a in these cells (**B**) were evaluated in basal conditions (gray lines) or after treatment with IL-15 for 48 h (black lines) after co-culture (1:1) for 1 h with Vero E6 cells infected with pseudotyped SARS-CoV-2. Each dot corresponds to the mean of data, and vertical lines represent SEM. Statistical significance between groups was calculated using Mann–Whitney U-test, and within groups, Wilcoxon U-test was used.

**Figure 6 ijerph-20-01947-f006:**
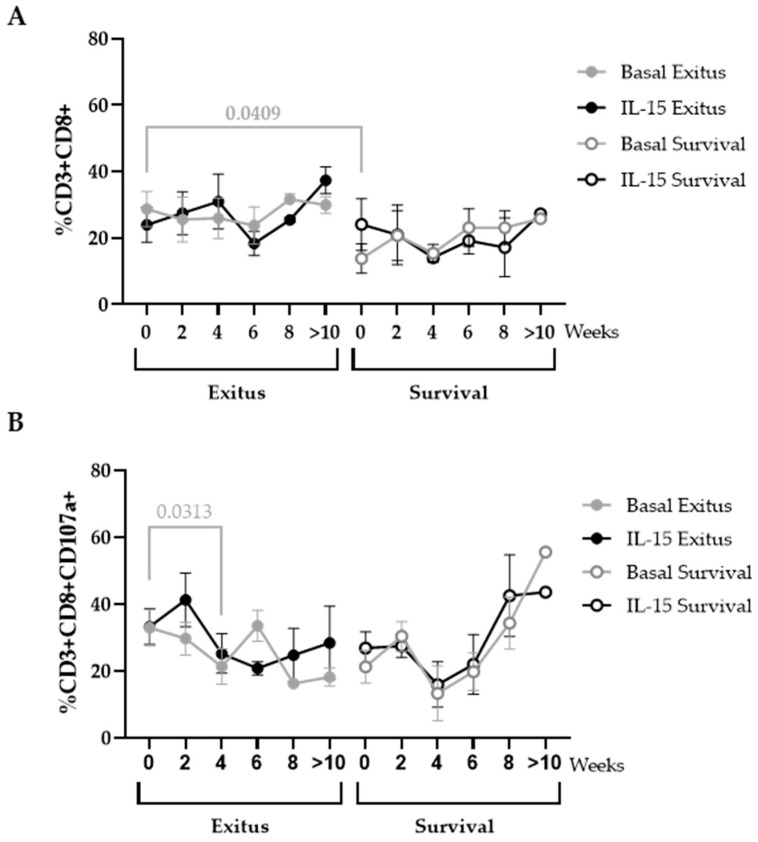
Levels of CD8+ T cells in PBMCs from individuals hospitalized with critical COVID-19. (**A**) Total levels of CD8+ T cells (CD3+CD8+) and (**B**) expression of the degranulation marker CD107a in these cells was evaluated in PBMCs in basal conditions (gray lines) or after treatment with IL-15 for 48 h (black lines) after co-culture (1:1) with Vero E6 cells infected with pseudotyped SARS-CoV-2 for 1 h. Each dot corresponds to the mean of data, and vertical lines represent SEM. Statistical significance between groups was calculated using Mann–Whitney U-test, and within groups, Wilcoxon U-test was used.

**Figure 7 ijerph-20-01947-f007:**
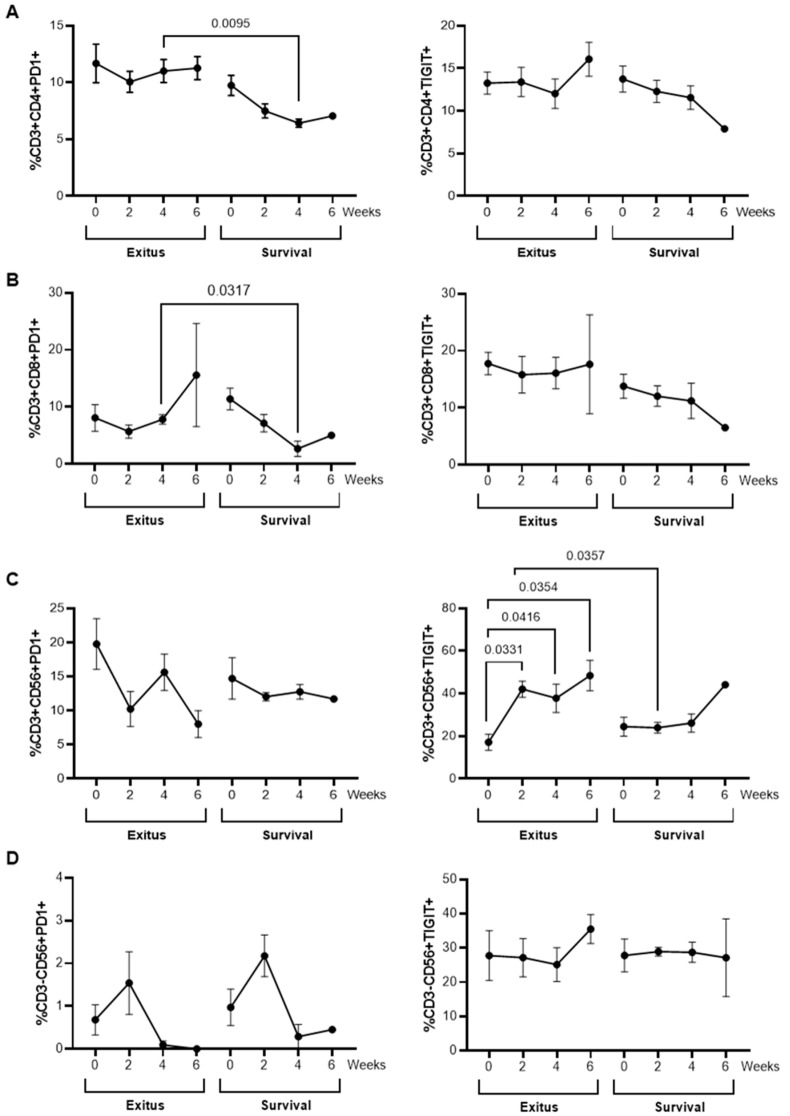
Expression levels of exhaustion markers in PBMCs from individuals hospitalized with critical COVID-19. Expression levels of PD-1 and TIGIT in CD4+ T cells (**A**), CD8+ T cells (**B**), NKT cells (**C**), and NK cells (**D**). Each dot corresponds to the mean of data, and vertical lines represent SEM. Statistical significance between groups was calculated using Mann–Whitney U-test, and within groups, ANOVA was used.

**Table 1 ijerph-20-01947-t001:** Demographic and clinical data of all participants from the Exitus group and the Survival group who were recruited for this study.

	Exitus (*n* = 13)	Survival (*n* = 10)
Demographic characteristics		
Age, median (IQR)	65.0 (62.0–69.0)	63.0 (59.0–68.5)
Sex: male, *n* (%)	11 (84.6)	6 (60.0)
Hospital stay		
Hospitalization days, median (IQR)	59.0 (36.0–100.0)	73.0 (44.5–90.0)
Days in ICU (IQR)	49.0 (25.5–82.0)	44.0 (25.0–66.5)
Comorbidities		
One or more comorbidity, *n* (%)	9 (69.23)	6 (60.0)
DM, *n* (%)	4 (30.8)	2 (20.0)
DL, *n* (%)	6 (54.5)	6 (60.0)
HT, *n* (%)	5 (41.7)	4 (40.0)
Treatments		
Corticoids, *n* (%)	10 (76.9)	9 (90.0)
Antibiotics, *n* (%)	10 (76.9)	9 (90.0)
Anticoagulants, *n* (%)	10 (76.9)	9 (90.0)
Tocilizumab, *n* (%)	3 (23.1)	1 (10.0)
Remdesivir, *n* (%)	1 (7.7)	1 (10.0)
Signs and symptoms		
Pneumonia, *n* (%)	13 (100)	10 (100)
Cough and expectoration, *n* (%)	11 (84.6)	9 (90.0)
Dyspnea, *n* (%)	11 (84.6)	9 (90.0)
Fever, *n* (%)	12 (92.3)	8 (80.0)
Blood biochemistry data at baseline		
CRP (mg/L) (IQR)	108.0 (30.2–202.0)	123.4 (58.1–172.6)
LDH (U/L) (IQR)	423.5 (319.8–535.8)	388.0 (365.0–470.0)
Lymphocytes (cells/μL) (IQR)	860.0 (620.0–1090.0)	1040.0 (930.0–1500.0)
Monocytes (cells/μL) (IQR)	490.0 (290.0–700.0)	670.0 (462.5–770.0)
Platelets (10^3^ cells/μL) (IQR)	191.0 (142.0–277.0)	284.0 (182.5–450.7)
Fibrinogen (mg/dL) (IQR)	740.0 (608.8–740.0)	740.0 (601.0–740.0)
Ventilation		
NIV, *n* (%)	1 (7.7)	0 (0)
IV, *n* (%)	12 (92.3)	10 (100.0)
Reservoir, *n* (%)	1 (7.7)	2 (20.0)
Nasal glasses, *n* (%)	1 (7.7)	5 (50.0)

CRP: C-reactive protein; DL: dyslipidemia; DM: diabetes mellitus; HT: hypertension; ICU: intensive care unit; IQR: interquartile range; IV: invasive ventilation; LDH: lactate dehydrogenase; NIV: non-invasive ventilation; SD: standard deviation.

## Data Availability

The original contributions presented in the study are included in the article. Further inquiries can be directed to the corresponding authors.

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
