# Peer review of "Sustained Cytotoxic Response of Peripheral Blood Mononuclear Cells from Unvaccinated Individuals Admitted to the ICU Due to Critical COVID-19 Is Essential to Avoid a Fatal Outcome"

_ijerph, 2023, doi:10.3390/ijerph20031947_

Round 1

Reviewer 1 Report

Casado and coauthors report the PBMC cytotoxicity studies of unvaccinated Covid-19 patients admitted to ICU. Total of 23 patients (13 exitus and 10 survival individuals) were recruited in this study and no significant differences between groups were found in sociodemographic, clinical, or biochemical data that may influence the fatal outcome. These individuals did not show lymphopenia or SARS-CoV-2 RNAemia. The key difference between survival groups and exitus groups is that the former PBMC shows significant higher cytotoxicity and positive response from IL-15 compared to the exitus group. The authors concluded that sustained, efficient cytotoxic activity is essential for survival during hospitalization due to severe COVID-19. 

The overall study is interesting and technically sound. However one minor comment is listed below.

On page 7, Figure 3. Justification or additional information is needed to assert that caspase-3 activity hence cytotoxicity measured was due to caspase-3 in Vero-E6 cell instead of the caspase-3 in co-cultured PBMC.

Reviewer 2 Report

The work by Casado-Fernandez et al explores the cytotoxic activity of NK and CD8 T cells from PBMCs in unvaccinated patients on COVID-19 disease, hospitalized until their exitus or discharge from hospital. These cells were colected during their hospitalization. The work is well-performed and written. The work is relevant for clinicians and the scientific and general audience.

Some aspects can be addressed by authors to support their work:

Healthy controls can be included in the survival and cytotoxic experiments.

Phenotypic analysis of exhaustion markers can be performed in CD8 T cells.

Determination of the levels of expression of IL-15 receptor are required.

Please, include your data on serum determination of RNA from SARS-CoV-2 (as Suppl. Material, if preferred).

Otherwise, the work is acceptable for publication.

Round 2

Reviewer 2 Report

I commend the authors on the work performed revising the manuscript.

Author Response

We would like to thank Reviewer 2 for the kind assessment of our work.